# Robustness of Classical and Quantum Inspired Architectures Against Structured Corruptions in Vision Tasks

**Abstract.** Robust performance under distributional shifts and noisy inputs is critical for real world deployment of machine learning models. While Convolutional Neural Networks (CNNs) remain the foundation of vision models, they are notoriously sensitive to corruptions in the input data, such as sensor noise, partial occlusions, or bit level errors. Motivated by the growing interest in quantum machine learning, we investigate whether quantum inspired architectural inductive biases can confer greater resilience to such perturbations.

We conduct a systematic evaluation of three neural architectures Classical CNN, Quantum Convolutional Neural Network (QCNN), and Quantum Multi Head Attention (QMHA) on the CIFAR 10 dataset under a diverse set of corruption regimes. Specifically, we test inference time robustness against Gaussian noise, salt and pepper corruption, Fourier masking, stripe noise, block masking, and bit flip noise, without any retraining or augmentation.

Our results demonstrate that while CNNs degrade significantly under most corruptions, quantum inspired architectures, particularly QMHA, exhibit improved robustness in multiple scenarios. These findings highlight the potential of quantum informed designs in developing resilient vision models and suggest promising directions for future hybrid quantum classical architectures in real world deployment settings.

## 1 Introduction

Deep learning has achieved state of the art performance in a wide array of computer vision tasks, ranging from image classification to object detection and segmentation. However, a critical limitation of modern neural networks is their fragility to corrupted, noisy, or adversarially perturbed inputs. Even minor distortions such as Gaussian blur, pixel masking, or bit flips can cause significant performance degradation. This vulnerability poses major challenges in deploying these models in safety critical or real world applications, such as autonomous driving, medical imaging, or robotics.

Recent work has explored data augmentation, adversarial training, and input preprocessing as means to improve robustness. However, these methods often increase computational cost and may generalize poorly to unseen corruption types. An emerging alternative is to redesign the model architecture itself to inherently withstand noise without relying on retraining or post hoc defenses.

In this paper, we explore the robustness properties of *quantum inspired neural architectures*, which integrate principles from quantum computing into classical

models to induce novel inductive biases. These principles include entanglement like feature mixing, probabilistic measurement via expectation values, and circuit based computation. Such inductive biases may enhance the model's ability to encode global structure, suppress irrelevant noise, and generalize under perturbations.

We evaluate three architectures:

- **CNN:** A standard convolutional neural network with ReLU and pooling layers.
- **QCNN:** A variant of CNN with quantum inspired latent projections and dropout like mechanisms.
- **QMHA:** A novel architecture combining CNN encoders with multi head quantum circuit simulations, where each head computes expectation values from parameterized quantum transformations.

Our contribution is a unified empirical study that tests these models across six distinct corruption types, each representing a different class of structured or unstructured noise. Crucially, all models are trained solely on clean data to isolate architectural robustness from training time exposure. We show that quantum inspired models particularly QMHA exhibit notable resilience across multiple corruption scenarios, supporting their relevance for robust machine learning in noisy environments.

## 2    Related Work

### 2.1    Robustness Benchmarks

Robustness in computer vision has become a growing concern, with datasets such as CIFAR 10 C [1] and ImageNet C providing standardized corruption benchmarks. These benchmarks evaluate model performance under corruptions such as noise, blur, compression, and weather artifacts. While many defenses rely on data augmentation, adversarial training, or test time adaptation, recent studies emphasize the importance of architectural robustness—that is, models that are resilient by design, even without retraining.

### 2.2    Quantum Inspired Neural Networks

Quantum Machine Learning (QML) explores using quantum computing for learning tasks, while quantum inspired neural networks aim to simulate aspects of quantum computation using classical hardware. Models such as the Quantum Convolutional Neural Network (QCNN) [2] mimic quantum feature maps and hierarchical entanglement, while attention based variants such as Quantum Multi Head Attention (QMHA) simulate parallel quantum circuits to compute attention via expectation values. Prior work has focused on performance and expressivity; here, we study their robustness.

## 2.3   Model Architectures

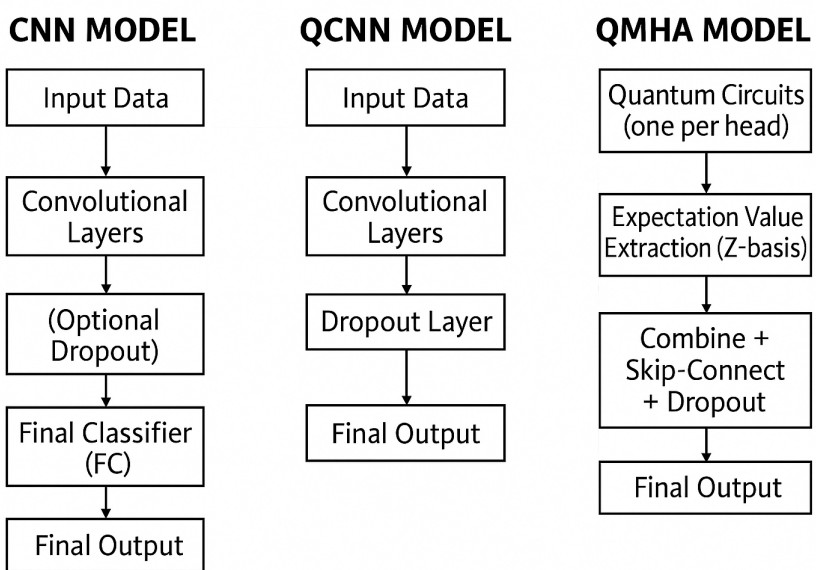

**Fig. 1.** Comparison of the three neural architectures evaluated in this study. The CNN model uses standard convolutional layers followed by an optional dropout and a final fully connected classifier. The QCNN model extends this by integrating a quantum inspired dropout mechanism and latent projection layers to simulate quantum feature encoding. The QMHA model replaces traditional convolutional processing with parallel quantum inspired attention heads, each simulating a parameterized quantum circuit and extracting Z basis expectation values. Outputs from each head are aggregated via skip connections and regularized using dropout before classification.

To isolate architectural robustness, all three models are trained on clean CIFAR 10 images using identical optimization settings. The primary difference lies in their architectural choices, especially the integration of quantum inspired inductive biases.

*CNN:* The classical baseline is a lightweight convolutional neural network comprising two convolutional layers with ReLU activations and max pooling, optionally followed by a dropout layer. The resulting features are flattened and passed through a linear classifier. This architecture represents a standard approach in low capacity vision models and serves as a reference for robustness degradation.

*QCNN:* The Quantum Convolutional Neural Network (QCNN) extends the classical CNN by incorporating two key enhancements inspired by quantum computing principles. First, a latent projection module is introduced after the convolutional backbone, simulating a quantum feature map by applying a high dimensional nonlinear embedding. Second, a dropout mechanism is employed that probabilistically suppresses feature channels in a pattern inspired by quantum decoherence. These modifications aim to introduce redundancy and disentanglement in the latent space, potentially improving robustness to structured noise.

*QMHA:* The Quantum Multi Head Attention (QMHA) architecture replaces the convolutional feature extractor with a quantum inspired attention mechanism. The encoder consists of a shallow CNN that projects inputs to a latent representation, which is then processed by $H$ parallel attention heads. Each head simulates a parameterized quantum circuit $U_i(x)$, and the output is computed as an expectation value in the Z basis:

$$z_i = \langle 0|U_i(x)^\dagger Z U_i(x)|0\rangle$$

The expectation values from all heads are concatenated and passed through a fusion layer with residual connections and dropout before being classified. This architecture introduces an implicit ensemble of disentangled feature paths, inspired by quantum entanglement, and is hypothesized to confer robustness by enabling the model to extract invariant representations under noise.

These models were designed to reflect a progression from purely classical architectures to increasingly quantum inspired designs, allowing us to directly assess the contribution of quantum inductive biases to robustness under various corruptions.

### 2.4 Noise Types

To evaluate the intrinsic robustness of each architecture, we apply a diverse set of structured and unstructured noise corruptions to the CIFAR 10 test dataset. These noise types were chosen to reflect common challenges in real world vision systems, including sensor degradation, transmission errors, and environmental distortions. Noise is applied post training, during inference only, without any retraining or augmentation, ensuring that robustness is purely architectural.

- **Gaussian noise:** Additive zero mean Gaussian noise with varying standard deviation $\sigma \in [0.0, 1.0]$, simulating camera sensor noise or low light environments.
- **Salt and pepper noise:** A portion of image pixels are randomly set to either 0 (black) or 1 (white), modeling binary sensor corruption and bit level faults.
- **Fourier masking:** Central frequency components in the Fourier domain are suppressed, reducing structural and low frequency information. This simulates transmission artifacts and lossy compression effects.

- **Stripe noise:** Random vertical columns are zeroed out, producing periodic occlusion patterns similar to column sensor failure or interlaced interference.
- **Block noise:** Square regions of the image are occluded with black patches, mimicking real world occlusions such as raindrops, dust, or partial visibility.
- **Bit flip noise:** Random pixel values are inverted, simulating low level hardware faults and memory corruption at the bit level, particularly relevant in embedded and edge computing scenarios.

Each corruption type is applied at multiple severity levels, and models are evaluated based on classification accuracy across these perturbations.

### 2.5   Training and Evaluation

All models are trained on clean CIFAR 10 images for 50 epochs using the Adam optimizer, with a learning rate of $1 \times 10^{-3}$ and a batch size of 32. To ensure fair comparison, all training hyperparameters are kept constant across models.

Importantly, no noise augmentation is used during training—noise is applied exclusively at test time. This ensures that the observed performance degradation stems from architectural robustness rather than exposure to corrupted samples during optimization.

Model evaluation is performed using top-1 classification accuracy on the corrupted test set. For each noise type, we report accuracy at two severity levels (e.g., $\sigma = 0.3$ and $\sigma = 0.5$ for Gaussian noise), along with accuracy under clean (unperturbed) conditions. No finetuning or postprocessing is performed.

## 3   Results

**Table 1.** Accuracy (%) on clean and corrupted CIFAR-10 test data. Abbreviations: g = Gaussian, s = Salt and Pepper.

| Model | Clean | g0.3 | g0.5 | g0.7 | s0.3 | s0.5 | Fourier | Bitflip |
|-------|-------|------|------|------|------|------|---------|---------|
| CNN | 57.46 | 24.60 | 15.88 | 13.09 | 19.24 | 14.06 | 22.39 | 54.85 |
| QMHA | 55.26 | 31.37 | 18.07 | 14.24 | 23.81 | 14.68 | 18.40 | 54.88 |
| QCNN | 57.94 | 24.97 | 15.09 | 12.58 | 19.17 | 12.75 | 19.27 | 56.52 |

*Observations:*

- **QMHA** shows superior robustness under Gaussian and salt and pepper noise, outperforming both CNN and QCNN by a notable margin at medium noise levels ($\sigma = 0.5$).
- **QCNN** slightly edges out the others under bit-flip noise, suggesting that its latent projection layer introduces redundancy beneficial for binary level perturbations.

    – **CNN** suffers significant accuracy drops across nearly all corruption types, especially in Fourier masked and block like scenarios where its local feature detectors become unreliable.

**Table 2.** Accuracy (%) on clean and corrupted CIFAR-10 test data with dropout.

| Model | Clean | g0.5 | g0.7 | g1.0 | s0.5 | s0.7 | Fourier | Bitflip |
|---|---|---|---|---|---|---|---|---|
| CNN | 54.51 | 15.07 | 13.10 | 10.65 | 12.36 | 10.62 | 16.08 | 53.88 |
| QMHA+DO | 54.90 | 17.26 | 16.01 | 12.24 | 14.70 | 11.82 | 23.36 | 54.06 |
| QCNN+DO | 56.30 | 16.51 | 14.34 | 11.50 | 13.31 | 11.41 | 20.01 | 55.07 |

*Additional Observations (with Dropout):*

– **QMHA+DO** achieves the best accuracy under Fourier corruption (23.36%), outperforming all other configurations, including the vanilla QMHA and CNN.
– Dropout improves robustness for both QMHA and QCNN models under Gaussian noise ($\sigma = 0.5$ and 0.7), though the gain diminishes at extreme levels ($\sigma = 1.0$).
– **QCNN+DO** continues to perform strongly under bit-flip corruption, maintaining high resilience even after dropout is introduced.
– While dropout does not dramatically improve salt and pepper robustness beyond QMHA's baseline, it does offer small consistent improvements and regularizes performance across moderate noise regimes.

## 4 Discussion

The results indicate that quantum inspired inductive biases can offer tangible robustness benefits in vision tasks without requiring noise specific training. The QMHA architecture, in particular, maintains stronger performance under unstructured and semi structured corruption types such as Gaussian and salt and pepper noise. This may stem from its use of disentangled attention heads and expectation value based feature extraction, which introduce implicit regularization and redundancy.

Interestingly, the QCNN's resilience under bit flip corruption implies that its latent projection encodes information in a way that tolerates binary level noise, possibly due to redundancy in its intermediate representations. This suggests that even relatively minor architectural changes inspired by quantum processing can improve robustness in specific scenarios.

However, no model demonstrated universal robustness across all corruption types. In fact, QMHA underperforms CNN under Fourier noise, indicating that high frequency or global distortions still challenge quantum inspired structures.

This emphasizes the importance of evaluating multiple corruption types and severity levels when assessing robustness.

## Limitations

While QMHA exhibits promising robustness, it is not universally superior. Under high-frequency corruptions such as Fourier masking, it performs worse than the CNN baseline. Additionally, quantum-inspired models incur additional computational complexity, especially with increasing attention heads or deeper circuit analogs. Future work must also validate robustness on larger, real-world datasets and resource-constrained deployments.

## 5 Conclusion

In this study, we compared the robustness of classical and quantum inspired architectures CNN, QCNN, and QMHA, on the CIFAR 10 dataset subjected to a diverse set of noise corruptions. All models were trained exclusively on clean data, isolating architectural robustness from data driven adaptation.

Our findings demonstrate that quantum inspired architectures, particularly QMHA, offer promising robustness advantages under common corruption types, including Gaussian and salt and pepper noise. These results suggest that incorporating quantum design principles such as circuit inspired attention, expectation value extraction, and entangled feature paths can enhance the resilience of deep learning models in adverse environments.

Future work may extend this analysis to:

- Larger and higher resolution datasets such as CIFAR 100 and ImageNet.
- Real world corruptions (e.g., weather, motion blur, lens distortion).
- Hybrid training regimes that combine robustness with noise aware fine tuning.

Ultimately, this study supports the growing interest in quantum inspired neural architectures not just for performance, but also for their potential to improve reliability and generalization in challenging settings.

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
