# OpenReview forum: "Robustness of Classical and Quantum Inspired Architectures Against Structured Corruptions in Vision Tasks"
_purdue.edu/Purdue_University/PQAI/2025/Symposium — PQAI 2025 Oral_

### Official Review · Reviewer_xdHd · 2025-07-24
**Nice results pointing to the robustness of quantum models**

**Rating:** 7
**Confidence:** 4

**Review:**

The paper investigates the robustness of various quantum learning models under different types of noise, using the CIFAR-10 dataset as a benchmark for vision tasks. Specifically, it evaluates three neural architectures: a classical Convolutional Neural Network (CNN), a Quantum Convolutional Neural Network (QCNN), and a Quantum Multi-Head Attention model.

The authors present numerical results suggesting that certain quantum architectures may exhibit greater robustness to structural corruption in the CIFAR-10 dataset. This finding points to a potential advantage of quantum models over classical counterparts in handling specific types of noise. This is an interesting result.

However, the scope of the study appears limited. The evaluation is confined to a single dataset, which raises concerns about the generalizability of the results.  Moreover, the observed performance improvements of quantum models over classical ones are relatively small.  In light of this, the practical significance of adopting quantum models for such tasks remains unclear. The marginal gains reported may not justify the added complexity and resource demands associated with quantum implementations, especially in the absence of broader empirical validation.  Despite that I think the paper adds values to the community.

---

### Decision · Program_Chairs · 2025-07-29

Accept (Oral)